# Mobile app for prolonged grief among bereaved parents: study protocol for a randomised controlled trial

Rakel Eklund  ,[1] Maarten C Eisma,[2] Paul A Boelen,[3,4] Filip K Arnberg,[1] Josefin Sveen[1]

¹Department of Neuroscience, Uppsala University, Uppsala, Sweden
²Department of Clinical Psychology and Experimental Psychopathology, University of Groningen, Groningen, The Netherlands
³Department of Clinical Psychology, Faculty of Social Sciences, Utrecht University, Utrecht, The Netherlands
⁴ARQ National Psychotrauma Centre, Diemen, The Netherlands

**Correspondence to**
Dr Rakel Eklund;
rakel.eklund@neuro.uu.se

## ABSTRACT

**Introduction** Bereaved parents, who have lost a child, have an elevated risk to develop mental health problems, yet, few studies have evaluated the effect of psychosocial interventions developed for bereaved parents. Cognitive–behavioural therapy (CBT), both face to face or digitally delivered, has shown to be an effective intervention for prolonged grief symptoms. Self-help mobile apps offer various advantages and studies show improved mental health after app interventions. No app has yet been evaluated targeting prolonged grief in bereaved parents. Therefore, the aim of this planned study is to develop and examine the effectiveness of a CBT-based mobile app, called *My Grief,* in reducing symptoms of prolonged grief, as well as other psychological symptoms, in bereaved parents. Another aim is to assess users' experiences and adverse events of *My Grief*.

**Methods and analysis** We will conduct a two-armed randomised waitlist-controlled trial. Parents living in Sweden, who lost a child between one and ten years ago, with elevated symptoms of prolonged grief, will be recruited to participate in the trial. The content of *My Grief* covers four main domains (Learn; Self-monitoring; Exercises; Get support) and builds on principles of CBT and the proven-effective *PTSD Coach* app. Participants in the intervention group will fill out online questionnaires at baseline and at 3, 6 and 12 months follow-ups, and the waitlist-controls at baseline and at 3 months. The primary outcome will be prolonged grief symptoms at the 3 months follow-up. Secondary outcomes are post-traumatic stress and depression symptoms, quality of life and cognitive behavioural variables (ie, avoidance, rumination, negative cognitions).

**Ethics and dissemination** Ethical approval has been received from the Swedish Ethical Review Authority (project no. 2021-00770). If the app is shown to be effective, the app will be made publicly accessible on app stores, so that it can benefit other bereaved parents.

**Trial registration number** NCT04552717.

## Strengths and limitations of this study

► This is the first study to examine whether access of a self-help app can have beneficial effect on bereaved parent's mental health and quality of life.

► This study will examine the effect of the app on cognitive behavioural processes proposed to underlie the development of prolonged grief.

► Generalisability of findings from this study may be limited as parents who want to participate in such study may experience fewer barriers to talk about the loss and seek help for their grief.

► This study includes parents who have lost a child, hence the findings may not be generalisable to other types of loss.

► This study uses self-report questionnaires, hence we do not establish formal diagnosis of prolonged grief disorder.

## INTRODUCTION

The death of one's child can be a traumatic life experience and bereaved parents are at increased risk of developing both mental and physical health problems.[1–6]

While most bereaved individuals adjust to the loss of a loved one without professional intervention, a significant minority experience a persistent, severe and disabling grief reaction, called prolonged grief. Prolonged Grief Disorder (PGD), a diagnosis characterised by such severe grief responses, was recently included in the International Classification of Disorders eleventh edition.[7] A similar version of PGD is scheduled to be included in the text revision of fifth edition of the Diagnostic and Statistical Manual of Mental Disorders (DSM-5-TR; for a commentary see Boelen *et al*[8]). Across both disorders, core symptoms are persistent and pervasive yearning for the deceased, persistent and pervasive cognitive preoccupation with the deceased, as well as accessory symptoms presumed indicative of intense emotional pain, such as difficulty accepting the loss, a feeling that one has lost a part of one's self, and difficulty in engaging in social activities. Such severe grief reactions often co-occur with, yet are distinct from, disorders such as depression and post-traumatic stress,[9] and relate to suicidal ideation,[10] as well as reductions of quality of life.[11]

Bereaved parents are at high risk to develop prolonged grief.[12–14] For example, 16% of cancer bereaved parents may have reactions indicative of PGD.[15] Yet, there is a lack of studies evaluating the effect of psychosocial interventions for bereaved parents.[16 17] Thus, the development and evaluation of accessible interventions to prevent and reduce negative mental health consequences after child loss appears clinically important.

A growing body of evidence shows that cognitive–behavioural therapy (CBT) is effective in treating prolonged grief symptoms when delivered face to face[18–21] or via internet.[22–24] Briefly, the cognitive behavioural model of prolonged grief proposes that three core processes explain the occurrence and persistence of grief symptoms: insufficient integration of the loss with existing autobiographical knowledge; negative global beliefs about oneself, the world, and the future and catastrophic misinterpretations of grief symptoms; and anxious avoidance (ie, cognitive and overt avoidance of reminders of the loss) and depressive avoidance (ie, avoidance of social, occupational and recreational activities, and behavioural withdrawal).[25] Accordingly, CBT treatments of prolonged grief typically includes creating a coherent, meaningful autobiographical narrative about the loss; challenging negative beliefs and catastrophic misinterpretations through cognitive restructuring; gradually confronting persons in vitro or in vivo with avoided aspects of the loss (eg, places, objects, memories) through exposure techniques; and/or helping people to set new life-goals and engage in new, meaningful activities.[26] Increasingly, researchers are also focused on examining the efficacy of other, complementary therapeutic techniques, which may be helpful in reaching these treatment goals, such as mindfulness-based therapy and relaxation.[27–29]

Mobile apps offer additional advantage over face-to-face and web-based interventions because of their availability, accessibility, the immediate support they can provide, and their anonymity (eg, reducing barriers to seeking help for grief), low costs and possible tailoring to the user (eg, the user are in control over when and how to use the app).[30] Most individuals own a smartphone, regardless of socio-economic background or living conditions, and many spend 2–5 hours a day using their smartphone.[31] Moreover, people with mental health issues report daily use of smartphones to connect to other people and search for health-related information.[32 33] Hence, mobile apps have a great potential to be used as self-help interventions for mental health problems. Indeed, various mobile apps for mental health have shown to be effective.[34 35] Treatment trials have shown that mobile apps can decrease a variety of psychological symptoms, including post-traumatic stress[36 37] but also depression,[38] anxiety[39] and substance use (eg, tobacco, alcohol and drugs).[40]

Despite the availability of a large number of mobile apps in app stores that claim to improve mental health, empirical evidence is lacking for most mobile applications.[30 34 35] To our knowledge, no evidence-based app to reduce grief for bereaved parents exists. Based on the existing knowledge, we believe an app targeting prolonged grief using elements of CBT could potentially be effective in improving mental health in bereaved parents.

The general goal of the present intervention study is to evaluate the effectiveness of an app for bereaved parents, which aims to facilitate the grieving process. The intervention includes access to a self-management mobile app, called *My Grief*, and is based on the cognitive behavioural conceptualisation framework of prolonged grief.[18 25] The intervention also includes other evidence-based techniques from CBT such as elements of mindfulness and deep muscle relaxation practices, as these techniques have shown promising effects on loss-related mental health for bereaved individuals.[27]

The main aim of this study is to examine the effectiveness of the app *My Grief* as compared with a waiting list comparison in reducing symptoms of prolonged grief in bereaved parents. The primary hypothesis is that the parents in the intervention group will report decreased levels of prolonged grief symptoms after having access to the app. A second aim is to examine the effect of the app on related mental health problems (post-traumatic stress symptoms, depression symptoms and quality of life) and cognitive behavioural variables putatively explaining the effect of CBT techniques (ie, grief avoidance, grief rumination and negative grief cognitions). The secondary hypotheses are that the parents in the intervention group will report improved mental health (ie, lower symptom levels of depression and post-traumatic stress, higher quality of life) and lower grief avoidance, grief rumination and negative grief-related cognitions. A third aim is to assess participants experience of using the app, for example, perceived satisfaction helpfulness and adverse events.

## METHODS AND ANALYSIS
### Design
We plan to conduct a two-armed parallel-group randomised waitlist-controlled trial comparing the effects of access to the *My Grief* for 3 months compared with waitlist. An external statistician will generate the allocation sequence: a random number table with equal allocation to access to *My Grief* or waitlist (1:1) with an unstratified, block design fixed at 20 allocations. The first author will upload the random number table, without reviewing it, into the Research Electronic Data capture (REDCap) randomisation tool prior to data collection. Randomisation will take place after the pre-assessment is completed. When both parents of a deceased child participate, the second parent taking part will be manually assigned to the same group as the first parent is assigned to through randomisation. Participants will be informed of group allocation immediately after randomisation (figure 1).

### Participants and procedures
Parents living in Sweden, who lost a child between 1 and 10 years ago, with elevated symptoms of prolonged grief, will be recruited to participate in the intervention.

The recruitment comprises two different procedures, (1) potential participants are identified using the Swedish

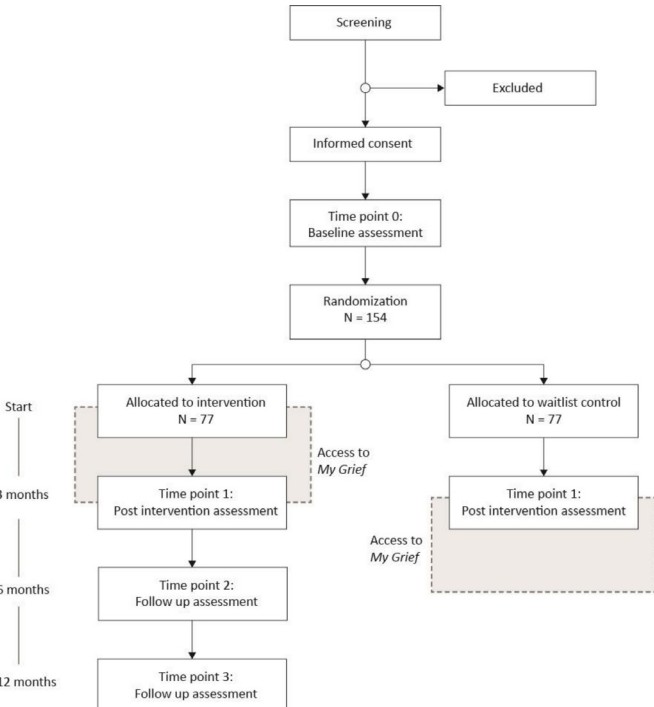

**Figure 1** Design of the randomised controlled trial.

Childhood Cancer Registry, the Cause of Death Registry, and the Swedish Population Register at the Swedish Tax Agency. By linking the Cause of Death Registry with the Swedish Childhood Cancer Registry, children diagnosed with a malignancy and who died 1–10 years previously will be identified. Thereafter, the children's parents are identified through the Swedish Population Register and will be sent a letter with an invitation to participate in the study. (2) Parents will also be recruited from non-profit organisations for bereaved parents who have lost a child (Vi Som Förlorat Barn and the Swedish Childhood Cancer foundation) via their social media sites and newsletters. People who are interested in participation, from both recruitment procedures, are directed to a website, www.minsorg.com, where they can access information about the study aims and procedures. On the website, they can sign up to the study and complete an online screening questionnaire which automatically direct eligible participants to an informed consent form and the pre-assessment online questionnaire, in REDcap.[41] After filling in the pre-assessment participants are randomised to either the intervention or waiting list group. Participants in the intervention group will receive a phone call from a researcher 1 week after the intervention started to ensure that the app is working properly. Participants in the intervention group are allowed to use the *My Grief* app for 3 months. The control group will be informed that they are on the waiting list and will be offered access to the *My Grief* app after 3 months (figure 1). Recruitment is planned to start August 2021.

### Eligibility criteria

Inclusion criteria for participation are: (1) being a parent of a child who has died between 1 and 10 years previously,

(2) having elevated symptom levels of prolonged grief (Prolonged Grief Disorder-13, (PG-13), cut-off >16), (3) understanding and speaking Swedish and (4) having access to a smartphone. Exclusion criteria are self-reported ongoing suicidal thoughts or psychosis, assessed with single items in the screening questionnaire.

### Sample size

A required sample size of 154 participants was calculated in G*Power V.3.1, based on the primary research question. We considered a moderate effect size to be clinically meaningful ($f$=0.25), within a repeated measure analysis of variance, a desired power of 0.80, $\alpha$=0.05, and an assumed strong association ($r$=0.50) between the preassessment and postassessment. The estimate includes an anticipated attrition rate of 16%.[42]

### Intervention

*My Grief* is a self-help app for smartphone users, compatible with both iOS and Android. The app builds on principles of CBT and a PGD online-treatment protocol.[43] It is based on a cognitive-behavioural conceptualisation of prolonged grief and proposes that three 'grief tasks' are critical in alleviating grief.[18 25] These tasks are (1) facing the loss and the pain that goes with it, (2) keeping confidence in yourself, others, life and future, and (3) engaging in helpful activities that promote adjustment to the new situation. Interventions used to accomplish these tasks include psychoeducation and normalisation of grief reactions, exposure to avoided aspects of the loss, and goal setting and behavioural activation. The structure of the app includes four main sections (figure 2) and is based on *PTSD Coach* app with permission from the Veterans' Affairs National Center for PTSD and Department of Defense's, Defense Health Agency, Connected Health (PTSD, Post Traumatic Stress Disorder).[36 44]

The four main sections in *My Grief* include:

### Learn

This section provides psychoeducation about grief. It covers what is normal- and prolonged grief; what commonly experienced thoughts, feelings and behaviours are after loss; the three grief tasks and coping with grief within the family. The latter topic includes tips on how to understand different family members' grieving processes, how to help each other, how to continue doing things together, how to support the remaining children in the family (ie, siblings to the deceased child) in their grief process, and how to talk to children about death. In each part, there are links to corresponding exercises in section 3.

### Self-monitoring

This section includes a rating scale (score 0–10) that users can employ to assess the daily intensity of their grief and the highest and lowest grief levels during the day. It also provides space to note what happened at that time. The section gives users both visual and written feedback from themselves as the results are presented together

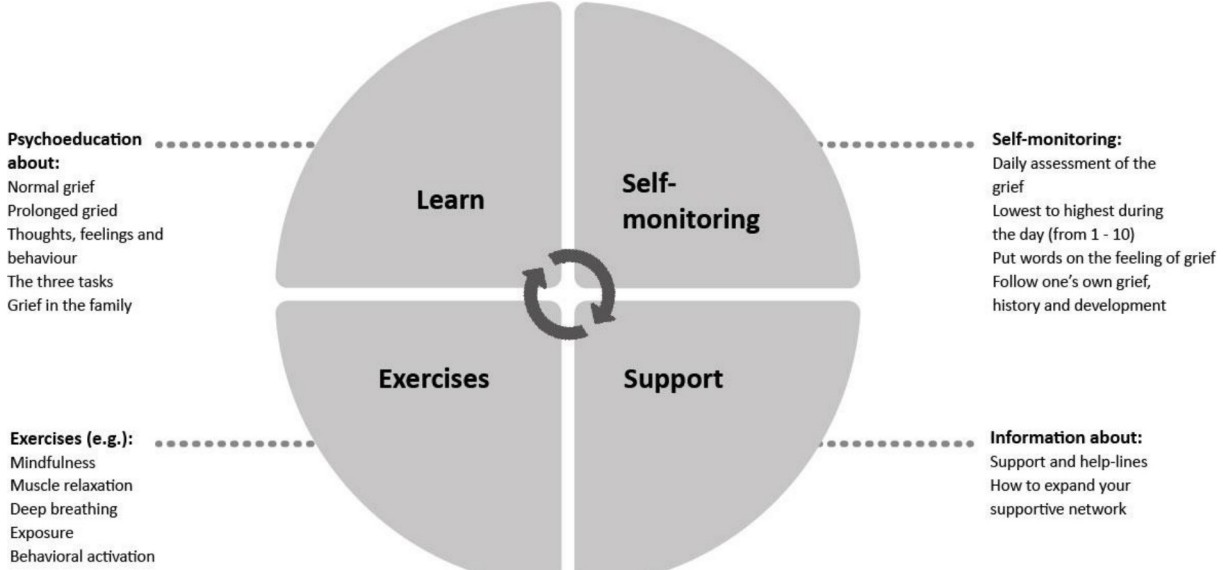

**Figure 2** Structure of the *My Grief* app.

with previous assessments. This enables users to monitor their grief intensity and symptoms over time, to recognise situations when grief levels are low or high, and to identify activities and internal and external stimuli associated with increases and decreases in grief; this helps them to learn more about one's own grief and to improve adaptive ways of coping. User can use the rating scale as many times as they want, whenever they want. Participants are encouraged to fill in the scale daily and they can choose to receive a daily reminder to fill in the scale.

## Exercises

This section provides the user with self-guided exercises. A majority of the exercises include recorded audio messages. For example, the user can choose to read or listen to instructions for a particular exercise. Some of the exercises were taken from the *PTSD Coach Sweden*,[45] including mindfulness exercises (eg, body scan, mindful breathing, learning to be aware of one's senses), progressive muscle relaxation, deep breathing, and stress reduction and positive psychology exercises. Other exercises were derived from CBT.[18 25] The writing exercise section includes tasks covering the three grief tasks in the CBT model. Three writing exercises, drawing from an internet-based exposure treatment,[43] are related to task one. In these writing exercises, participants expose themselves to the story of loss and are encouraged to confront implications of the irreversibility of the separation from the deceased. For example, in the exposure to the story of the loss the participants are instructed to write down what happened around the time of the death, in three steps, in order to confront feelings, thoughts, and situations which are difficult and emotionally painful. One exercise is related to grief task two and instructs participants to take a few minutes every day to think positive about the future and what they want for the future and to write this down in a diary. One exercise related to grief

task three, is based on behavioural activation principles. It consists of encouragement to plan a moment every day to engage in an activity that potentially elicits some pleasure or joy.

## Get support

This section provides the user with contact details to different support functions and help-lines (including phone numbers, chats, email addresses, and websites) for both the parent and his/her family members. These support functions or help-lines include the Childhood Cancer Foundation, a suicide prevention organisation, and different networks and non-profit organisations for parents who lost a child. It also includes psychoeducation regarding how the user can create and expand one's own support network throughout reaching out for help to family and friends. Participants can create a list of phone-numbers to family, friends and help-lines in this section as well.

## Data collection

Participants complete questionnaires online. Time points for data collected via online questionnaires at t0 (baseline), t1 (3 months), t2 (6 months, only intervention group) and t3 (12 months, only intervention group) (table 1; figure 1). If the participant has not completed the online questionnaire, two reminder emails will be sent out for each measurement time point.

## Sociodemographic and loss-related variables

A self-constructed questionnaire will be used to gather information about the participants' demographic information, such as age, gender, education, employment, the child's diagnosis (if applicable), cause of death and date of the child's death. These data will be used to describe the sample.

| Table 1 | Time points for measurements | | | | | |
|---|---|---|---|---|---|---|
| | | Before baseline | t0, baseline | t1, 3 months | t2, 6 months* | t3, 12 months* |
| Consent | | X | | | | |
| Screening | | X | | | | |
| Sociodemographic | | | X | | | |
| Prolonged grief (Prolonged Grief Disorder-13) | | | X | X | X | X |
| Post traumatic stress (Post-traumatic Stress Disorder Checklist-5) | | | X | X | X | X |
| Depression (Patient Health Questionnaire-9) | | | X | X | X | X |
| Quality of life (Brunnsviken Brief Quality of Life Inventory) | | | X | X | X | X |
| Grief avoidance (Depressive and Anxious Avoidance in Prolonged Grief Questionnaire) | | | X | X | X | X |
| Grief rumination (Utrecht Grief Rumination Scale) | | | X | X | X | X |
| Negative grief cognitions (Grief Cognition Questionnaire-18) | | | X | X | X | X |
| Users' experiences* | | | | X | | |

*Only intervention group.

## Primary outcome

Prolonged grief symptoms will be measured by 'PG-13'. It consists of 13 items including 11 items assessing cognitive, behavioural and emotional symptoms during the past month, rated on a 5-point scale (not at all—several times a day/overwhelmingly). It also contains two items on duration and impairment (yes/no). PG-13 is scored by summarising the 11 items assessing symptoms. The total score ranges from 11 to 55, with higher scores indicates more severe PGD.[46] The instrument is validated in a Swedish sample with bereaved parents and indicates satisfactory psychometric properties.[15]

## Secondary outcomes

PTSD symptoms will be measured with the PTSD Checklist for DSM-5 at baseline and follow-up. It consists of 20 items describing symptoms of post traumatic stress. Participants rate on a 5-point scale (not at all—extremely, 0–4) to what extent they experience symptoms. The total score ranges from 0 to 80, and a higher score indicates more symptoms of post-traumatic stress.[47] The Swedish version has shown satisfactory psychometric properties.[48]

Depression symptoms will be assessed with the Patient Health Questionnaire-9, which consists of nine items describing symptoms of depression. Participants rate to what extent they experience these symptoms during the last 2 weeks on a 4-point scale (not at all—nearly every day, 0–3). The total score ranges from 0 to 27, and a higher score indicates greater symptom severity.[49]

Quality of life will be measured with the Brunnsviken Brief Quality of Life Inventory. It consists of 12 items covering six different life domains. Participants rate on a 5-point scale (strongly disagree—strongly agree, 0–4), to what extent they agree with the statement. Total score ranges from 0 to 96. The instrument is validated in a Swedish sample.[50]

Grief-specific avoidance will be measured with the Depressive and Anxious Avoidance in Prolonged Grief Questionnaire (DAAPGQ). The questionnaire consists of nine items; five items measure depressive avoidance of activities and four items measures anxious avoidance of stimuli reminding of the loss. Participants rate on an 8-point scale (not at all true for me—completely true for me, 0–7) to what extent they experience the symptoms.[51] The total score ranges from 0 to 72. The DAAPGQ was translated into Swedish by two grief researchers (JS & RE) at the Department of Neuroscience, Uppsala University in 2020 following European Organisation for Research and Treatment of Cancer (EORTC) guidelines.[52]

Grief-specific rumination will be measured with the Utrecht Grief Rumination Scale. The instrument consists of 15 items measuring different aspects of grief rumination. The participants rate on a 5-point scale (never-very often, 1–5) how often they experienced certain thoughts over the last month.[53] Total scores range from 15 to 75 and generate an overall grief rumination score. The instrument is validated in a Swedish sample with bereaved parents and indicates satisfactory psychometric properties.[54]

Negative grief cognitions will be measured using a short version of Grief Cognitions Questionnaire (GCQ-18). It consists of 18 items, with statements regarding negative grief cognitions. Participants rate on a 6-point scale to what extent they agree with the statement (disagree strongly—agree strongly, 0–5). The total score ranges from 0 to 90[55] and a higher score indicates stronger endorsement of negative cognitions.[56] The GCQ was translated into Swedish by two grief researchers (JS and RE) at the Department of Neuroscience, Uppsala University in 2020 following EORTC guidelines.[52]

### Users' experiences: perceived satisfaction, helpfulness and adverse events

A questionnaire with 25 items (9 of them with free-text response) regarding the user experience of the app will be used after participants have had access to the app for 3 months. The questionnaire includes questions regarding for example the perceived helpfulness and benefits of the app, which sections and tools the participants used most commonly and why, and if any parts of the app could be improved. Adverse events are assessed through the question 'Did the use of the app have any negative consequences?' Participants who answered yes are asked to expand on this.

### Data management

Study data are managed using REDCap electronic data capture tools[41] hosted at Uppsala University. All data will be deidentified with a code number. The key to the codes will be stored separately, and is only available to members from the research group.

### Statistical methods and analysis

To examine intervention effects on the primary and secondary outcome variables, multiple regression models with the direct and interaction effect of condition ×time with the primary endpoint at 3 months follow-up will be performed. Missing data at follow-up will be handled under the assumption of missing at random. Data of all randomised participants will be included in the analyses (ie, intention-to-treat analysis). Cohen's $d$ and a 95% CI will be calculated to assess the within and between-group effect sizes at postintervention and follow-ups. Within group Cohen's $d$'s, the standardised mean difference between the pre-assessment and the post-assessment for each group, will be calculated.[57] Between groups Cohen's $d$'s will also be calculated by dividing the differences between change scores of both groups across time by the pooled SD of both groups at baseline.[58]

### Patient and public involvement

Two parents that lost their children were taking part in reading the content of the app and help out with translation of some questionnaires. The app has been tested by 10 cancer-bereaved parents during April–May 2021. The parents were asked to use *My Grief* for 4 weeks. After that, an in-depth assessment of the user experiences of the app was conducted via telephone, by a member of the research team with questions, for example, what it was like to use the app, how often they used it, which tools they used most commonly and why, and also whether they have suggestions to improve the app. The input and results of this testing lead to minor changes in the intervention.

### Ethics and dissemination

All participants will fill out their consent to participate before enrolment, and they will be informed that they can decline their participation at any time without giving a reason. The pilot study (project no. 2020-01704) and the randomised control trial (project no. 2021-00770) have received ethical approval from the Swedish Ethical Review Authority. Collected data will be handled confidentially, according to the European Union General Data Protection Regulation.

If the app is shown to be effective, the app will be made publicly accessible on app stores, so that it can benefit other bereaved people. Our findings will be presented to the Swedish Childhood Cancer foundation and other non-profit organisations for bereaved people. Colleagues will be informed about our findings during presentations at conferences and publications in scientific journals.

Taking part in this intervention could generate negative feelings and bring back negative memories, as it will elicit thoughts about the loss of the child. Any participant that reports experiencing harm or negative consequences as a result of their participation can, on their request, be referred to an appropriate healthcare provider. However, we anticipate that participants can also experience positive outcomes, such as learning more about their own grief, increases in self-awareness and improvements in mental health. Previous research shows that parents who lost a child to cancer valued participation in research as a positive experience despite extensive questionnaires of potentially sensitive character.[59] Previous research also shows that participants who have experienced trauma show no negative effects of using the similar app *PTSD Coach*, and indicate that the app is both acceptable and feasible to use.[42 44 45 60] Therefore, we believe that the benefits of participation in the study outweigh the risks.

## DISCUSSION

In this study protocol, we describe the development and planned evaluation of the self-management smartphone app *My Grief* for parents with prolonged grief symptoms after losing a child. This study will clarify whether the access of an app can have beneficial effects on bereaved parent's mental health, quality of life and on cognitive behavioural processes proposed to underlie the development of prolonged grief.

Offering a self-help app to bereaved parents in order to facilitate their grief process may have potential advantages, such as reducing barriers to seeking help, for example, finding it too painful to speak about the loss, difficult to find appropriate help, or the negative effects of mental health stigma on help-seeking.[13 61] The app is available free of charge to the users, and could be used in any geographic location and does not require visiting a healthcare organisation. However, the *My Grief* app is not a guided treatment as it does not include any personal feedback from healthcare professionals or the research group, which could be perceived as a limitation. Studies on the PTSD Coach app have had dropout rates below 25%[36 45] and an randomised controlled trial (RCT) on the *PTSD Coach* in Sweden that is not yet published (ClinicalTrials.gov, ID: NCT04094922) conducted by authors JS and FA has a dropout rate of 11%.

Generalisability of findings from this study may be limited for a number of reasons. Possibly, people who already received therapy or other social support prior to the loss may be more willing to participate in this study, because they may experience fewer barriers to seek help or talk about psychological problems.[13 61] Similarly, people who are more willing to use technology, such as mobile apps or internet-treatments, would probably be more willing to participate in an app-based study.[61–63] The generalisability of study outcomes may be affected in two ways by this selection bias. Tech-savvy users may appreciate the mode of delivery more which could have a positive impact on outcomes. Alternatively, those with more experience of technology could have higher standards and therefore higher demands of the app, which could affect their user experience and outcome in more negative ways. As seen in previous studies with the *PTSD Coach* app, some participants may experience difficulties with downloading or installing the app, or to remember to use it more than once during an ongoing intervention.[64 65] In the *My Grief* app, a daily reminder can be set up to remind them to rate their grief in the grief chart self-assessment. The research group will also make a phone call to each user, about 1 week from registration, to check if everything is working. We expect that these measures will increase participant involvement in using the app.

To conclude, this RCT will provide new insight to the effectiveness of a self-management app for prolonged grief in bereaved parents. If the app is found to be beneficial, it will provide a valuable addition to the bereavement support offered to bereaved parents in Sweden and may complement other psychological, medical and social interventions for prolonged grief. In addition, a next step could be to develop versions of the app for other groups of bereaved individuals and in other languages.

**Acknowledgements** Thanks to the Veterans' Affairs National Centre for PTSD and Department of Defence's DHA Connected Health for letting us base the *My Grief* app on the PTSD Coach source code. A special thanks to the two bereaved parents helping us with reading and commenting on the content in the app.

**Contributors** JS is principal investigator and grant holder. RE and JS are executive researchers and wrote the ethics proposals. RE, JS, MCE, FKA and PAB developed the study design. RE, JS and MCE developed the draft of the manuscript. FKA and PAB read, revised and approved the draft of the manuscript. All authors approved the final manuscript.

**Funding** This research was funded by the Swedish Childhood Cancer Fund (PR2018-0047; TJ2018-0002).

**Disclaimer** This funding source had no role in the design of this study and will not have any role during its execution, analyses, interpretation of the data, or decision to submit results.

**Competing interests** None declared.

**Patient and public involvement** Patients and/or the public were involved in the design, or conduct, or reporting, or dissemination plans of this research. Refer to the Methods section for further details.

**Patient consent for publication** Not applicable.

**Provenance and peer review** Not commissioned; externally peer reviewed.

**ORCID iD**
Rakel Eklund http://orcid.org/0000-0002-9396-9800

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
