## [Reviewer comments · BMJ Open]

ARTICLE DETAILS

TITLE (PROVISIONAL)	A Mobile App for Prolonged Grief among Bereaved Parents: Study Protocol for a Randomized Controlled Trial
AUTHORS	Eklund, Raket; Eisma, Maarten; Boelen, Paul; Arnberg, Filip; Sveen, Josefin

VERSION 1 – REVIEW

REVIEWER	Meiser-Stedman , Richard University of East Anglia
REVIEW RETURNED	30-Jun-2021

GENERAL COMMENTS	This protocol describes an important and compelling proposed randomised controlled trial of a mobile app for the treatment of prolonged grief in bereaved parents. The background to the present trial is well explained and the trial design is robust. I only have a few comments that I hope will be of use to the protocol authors. 1. Abstract, line 37 (and elsewhere): Might “Understanding” or “Psychoeducation” work better instead of “Learn”?2. Line 149 and following: The use of a random number generator as well as block randomization is confusing; a random number generator implies “simple randomization”, while randomization is a different method altogether for generating an allocation sequence. This section needs to be clearer. Do the authors mean block randomisation with block sizes that are varied at random? This would appear to be the ideal allocation strategy here.3. Line 176 – apologies if I have missed it, but I could not see the cut-off score on the prolonged grief measure that the authors are proposing to use for trial entry. It would be helpful to report this cut-off in advance.4. Line 318: The authors propose undertaking baseline comparisons (“T-tests and χ^2 tests will be used to compare groups on all loss-related and sociodemographic variables.”) My understanding is that this practice is discouraged (e.g. Senn S. Testing for baseline balance in clinical trials. Stat Med 1994; 13: 1715–26; Altman DG. Comparability of randomised groups. Statistician 1985; 34: 125–36; Assmann et al. 2000, Lancet, Subgroup analysis and other (mis)uses of baseline data in clinical trials)5. Line 329: Consider labelling this section “Patient and public involvement and piloting”?
--

REVIEWER	Hanna, Jeffrey University of Oxford, Department of Psychiatry
REVIEW RETURNED	24-Aug-2021

GENERAL COMMENTS	I would like to extend my congratulations to the authors for producing this comprehensive protocol that was clear and logical to follow. I recommend it suitable for publication in BMJ Open. I have one minor comment: Could the authors make it clear in the introduction section of the abstract this study is about when a child has died as often the term 'bereaved parents' can refer to when a co-parent has died. All the best with this study - I look forward to seeing this study progress.
---

VERSION 1 – AUTHOR RESPONSE

Dear Reviewers,

We are grateful for the work you have put into our manuscript and hope you will find it revised accordingly. We also want to inform you, that we decided to include parent's that lost a child, not only to cancer, but for other reasons as well. This decision was made in between time of submission (April 2021) and now. Therefore, we added some information regarding that in the manuscript as well. The manuscript therefore exceeds the word count with 176 words. In the table below, please find your comments/questions and the actions we have taken in revising our manuscript. Revisions are marked in *yellow* in the revised manuscript and referred to below by page in the new submitted version. Withdrawn text have not been retained as crossed-out text.

We look forward to hearing from you!

List of comments and actions taken:

Reviewer 1 comments	Answers from authors	Pages
Abstract, line 37 (and elsewhere): Might "Understanding" or "Psychoeducation" work better instead of "Learn"?	Thank you so much for your valuable review! We choose to keep the same wording as in PTSD Coach, the app that My Grief is based on and inspired by.	Abstract (and elsewhere)
Line 149 and following: The use of a random number generator as well as block randomization is confusing; a random number generator implies "simple randomization", while randomization is a different method altogether for generating an allocation sequence. This section needs to be clearer. Do the authors mean block randomisation with block sizes that are varied at random? This would appear to be the	Thank you for pointing this out, we have now revised this section to be clearer.	Page 7, line 149 and following.

ideal allocation strategy here.		
Line 176 – apologies if I have missed it, but I could not see the cut-off score on the prolonged brief measure that the authors are proposing to use for trial entry. It would be helpful to report this cut-off in advance.	The cut-off for trial entry has been added.	Page 8, line 184-185
Line 318: The authors propose undertaking baseline comparisons (“T-tests and χ^2 tests will be used to compare groups on all loss-related and sociodemographic variables.”) My understanding is that this practice is discouraged (e.g. Senn S. Testing for baseline balance in clinical trials. Stat Med 1994; 13: 1715–26; Altman DG. Comparability of randomised groups. Statistician 1985; 34: 125–36; Assmann et al. 2000, Lancet, Subgroup analysis and other (mis)uses of baseline data in clinical trials)	We have now removed the baseline comparison from the manuscript.	Page 14, line 327
Line 329: Consider labelling this section “Patient and public involvement and piloting”?	Thank you for that suggestion, we added “and piloting” to the heading at first, but then got back from the journal that we could not change this mandatory heading.	Page 15 line 338
Reviewer 2 comments	Answers from authors	Pages
Could the authors make it clear in the introduction section of the abstract this study is about when a child has died as often the term 'bereaved parents' can refer to when a co-parent has died.	Thank you so much for review of our manuscript! We added your input to the abstract.	Abstract